# What motivates general practitioners to change practice behaviour? A qualitative study of audit and feedback group sessions in Dutch general practice

Marije van Braak,[1] Mechteld Visser,[2] Marije Holtrop,[3] Ilona Statius Muller,[4] Jettie Bont,[2] Nynke van Dijk[2]

¹General Practice, Erasmus Medical Centre, Rotterdam, The Netherlands
²General Practice, Amsterdam University Medical Centre, Amsterdam, The Netherlands
³General Practitioners Holtrop and Sieben, Amsterdam, The Netherlands
⁴Ubbens and Statius Muller General Practitioners, Amsterdam, The Netherlands

**Correspondence to**
Marije van Braak;
m.vanbraak@erasmusmc.nl

## ABSTRACT

**Objectives** Adopting an attributional perspective, the current article investigates how audit and feedback group sessions contribute to general practitioners' (GPs) motivation to change their practice behaviour to improve care. We focus on the contributions of the audit and feedback itself (content) and the group discussion (process).

**Methods** Four focus groups, comprising a total of 39 participating Dutch GPs, discussed and compared audit and feedback of their practices. The focus groups were analysed thematically.

**Results** Audit and feedback contributed to GPs' motivation to change in two ways: by raising awareness about aspects of their current care practice and by providing indications of the possible impact of change. For these contributions to play out, the audit and feedback should be reliable and valid, specific, recent and recurrent and concern GPs' own practices or practices within their own influence sphere. Care behaviour attributed to external, uncontrollable or unstable causes would not induce change. The added value of the group is twofold as well: group discussion contributed to GPs' motivation to change by providing a frame of reference and by affording insights that participants would not have been able to achieve on their own.

**Conclusions** In audit and feedback group sessions, both audit and feedback information and group discussion can valuably contribute to GPs' motivation to change care practice behaviour. Peer interaction can positively contribute to explore alternative practices and avenues for improvement. Local or regional peer meetings would be beneficial in facilitating reflection and discussion. An important avenue for future studies is to explore the contribution of audit and feedback and small-group discussion to *actual* practice change.

## INTRODUCTION

In taking the Hippocratic oath, general practitioners – and other doctors as well – express their intention to treat patients to the best of their ability. Yet, care practices of general practitioners show substantial unintended variation.[1–3] Part of this diversity is induced

### Strengths and limitations of this study

► Framed within attribution theory, the study provides a novel perspective on audit and feedback.
► Focus group discussions on personal and comparative audit and feedback allowed us to tap into real-time reflective processes.
► Qualitative analysis of recorded interaction between general practitioners (GPs) allowed for detailed insight into the value of peer interaction in discussions on practice change.
► Voluntary GP participation may have resulted in a sample of participants with a special interest in audit and feedback and behaviour change.
► The study focused only on effects of audit and feedback and group discussion on intended, not actual, practice change.

by external practice or population factors, such as practice size.[2] Individual factors also play a role: general practitioners' knowledge, skills, experience, interests and preferences can induce between-practice variation. This variation might be related to differences in clinical judgement based on considerations of evidence, clinical experience and patient preferences,[4] but sometimes is not intended and may lead to lower patient care quality.

For general practitioners (GPs) to develop their professional practice, they should be aware of this unintended variation. Reflection on between-practice variation can lead to adjustments in professional care behaviour, eventually improving the quality of patient care.[5] Audit and feedback on GPs' performance can effectively improve professional practice, although under certain optimally-designed conditions and in the right context.[6] Several factors influencing the effectiveness of feedback have been researched. These include the level of feedback detail,[7] its timing[8 9] and the interactivity

of the feedback-giving process[4 5 9 10]. Based on expert interviews, systematic reviews and experience, Brehaut and colleagues suggest that practice feedback interventions can be optimised by, among others, linking it to established goals, providing feedback in more than one way, minimising extraneous cognitive load for feedback recipients, increasing the credibility of the data and preventing defensive reactions to feedback.[11] Additionally, recent research on group audit and feedback points at the added value of socially constructed learning activities in audit and feedback group sessions.[12]

The theoretical framing of audit and feedback research is diverse, ranging from feedback theories (eg, feedback intervention theory[13]) to psychological theories (eg, self-affirmation theory[14]) to implementation theories (eg, consolidated framework for implementation research[15]) to learning theories (eg, social learning theory[16]).[17] In our study, we explicitly focus on GPs' *motivation* to change. This perspective on audit and feedback has not been used before. Yet, motivation is essential to learning and change processes[18] and has been found to influence change behaviour of various physicians.[19–21] In a study on physicians' prescription behaviour, Wakefield *et al*[22] found that physicians who expressed a commitment to change following participation in a continuing medical education programme using interactive small groups were significantly more likely to change their targeted prescription practices in the following half year. Contradictory evidence, however, has also been reported.[23 24] Palmer *et al*[23] assessed the motivation of health professionals to change their practices and found that they improved on tasks for which they reported to have *limited* motivation. This counterintuitive finding can be explained by the significant *system* changes (in addition to individual changes) required to change practices for which professionals' motivation was high. Thus, the relation between motivation and care practice improvement may be less straightforward than originally anticipated.

The reported variability in the effect of motivation on actual care practice change can be explained by attributional processes that impact an individual's motivation to change.[25] For example, during audit and feedback meetings GPs might attribute practice variation to 'the system', particular patients, or fellow healthcare professionals (ie, hold others responsible for practice variation). In such a case, the GPs are not likely to be inclined to change their own professional practice behaviour.[11] Thus, GPs' attributions are significant in processes of general practice change and need careful consideration in the context of audit and feedback. In our study, therefore, we adopted an attributional perspective on audit and feedback.

In the Dutch context, audit and feedback have become increasingly important for GP professional development. Historically, GPs use pharmacological feedback from pharmacists and their electronic health records to improve prescribing behaviour – mainly in educational group sessions. In the last decade, GPs have started to use these sessions to discuss diagnostic procedures. At the same time, insurance companies have started to request from GPs information on quality indicators. As GPs expressed a need for audit and feedback sessions based on these quality indicators, the Dutch movement 'Optimale zorg-Dappere dokters' (Optimal care-Daring doctors) initiated an audit and feedback group intervention, aiming to encourage GPs' self-reflection and optimise care.

Group-based critical self-evaluation based on quality indicators is not standard in contemporary quality policies of the medical profession, but can be beneficial to stimulate self-reflection for optimal care.[12] Understanding how participants respond to the audit and feedback data will facilitate improvement of interventions. To this end, we qualitatively investigated how audit and feedback group sessions contribute to GPs' motivation to change their practice behaviour to improve care. We focused on GPs' attributions regarding both content and process of the audit and feedback group sessions by asking:

1. How does the audit and feedback itself contribute to GPs' motivation to change?
2. How does the group discussion contribute to GPs' motivation to change?

## THEORETICAL FRAMEWORK
The role of attributions in motivation and behaviour change is described in Weiner's attribution theory.[26 27] This theory indicates that humans 'have a tacit goal of understanding and mastering themselves and their environment' and 'establish cause-effect relationships for events in their lives'.[25] Occurring events lead to attribution, a process of (often subconsciously) seeking an explanation by hypothesising perceived personal and environmental causes (eg, ability, effort, luck, task difficulty, mood, health, other people, etc). These causes can be organised along three causal dimensions: locus (ie, internal or external to the individual), stability (ie, stable, fixed or unstable, likely to change) and controllability (ie, within or outside the individual's control). Based on the specific combination of values in each dimension, occurring events are interpreted as psychologically meaningful responses. As such, the individual's *interpretation* of a particular event determines their response to that event. Typically, bad luck is interpreted as external, unstable and uncontrollable – invoking no or possibly only a passive reaction from the individual; personal effort is internal, changeable and controllable – and can thus be influenced by an individual's actions; and innate skill is internal, largely fixed and uncontrollable – making it an unlikely subject of change.[25]

## METHODS
### Ethics
The Medical Ethics Review Committee of the Amsterdam University Medical Centre confirmed that the Medical Research Involving Human Subjects Act does not apply to

van Braak M, *et al. BMJ Open* 2019;**9**:e025286. doi:10.1136/bmjopen-2018-025286

this study and that an official approval of this study by the committee was not required (reference number W18_200 # 18.241).

## Patient and public involvement

No patient or public were involved. The development of the research question was informed by two practising GPs (MH, ISM), who also participated in the design of the study. The results will be disseminated to participants via an email informing GPs about the main results and focusing on issues of future audit and feedback implementation.

## Setting

The study was carried out in the region of Amsterdam. Dutch general practice plays an important role in the Dutch healthcare system, since specialist care in the Netherlands is only accessible on referral by a GP.[28] All Dutch citizens are registered at a GP-practice in their regional area. Patients visit their GP when faced with a medical problem (except in life-threatening situations). The GP collects and evaluates all relevant medical information. Consequent treatment decisions and referrals to a medical specialist are taken together with the patient. Costs of care are covered by healthcare insurances, which are compulsory for people who live or work in the Netherlands and which include at least basic healthcare.

Audit and feedback information is gathered and provided to GPs by Vektis, a Dutch centre for information and standardisation of health insurance. This information is based on declarations of healthcare costs to insurance companies. It provides data on indicators regarding practice population, consultations, interventions, prescription and referral rates compared with a standardised Dutch practice, corrected for age, gender, social-economical status and disease severity of the population.

## Data collection

We conducted focus group discussions[29 30] with GPs of four regional GP groups within the Amsterdam region. GP groups interested in this audit and feedback intervention could participate if their practices were in the Amsterdam region. General practitioners were approached and informed about the research purpose and participation practicalities via an information letter. Participating GPs signed informed consent prior to the discussions and sent the audit and feedback information that their practices received from Vektis (data from 2012 to 2014) to the research team for analysis. The focus group discussions were held between June 2016 and March 2017 at a participating GP's practice, lasted approximately 1.5 hours and were audio-taped for transcription purposes.

Focus group discussions were facilitated by a moderator (ND or MV). They guided the group through the data reports. Resembling Cooke *et al*'s intervention,[12] an aggregate comparative report of quality indicators (selected for their relevance by each focus group in a session preceding the audit and feedback session) was projected for everyone to see; actual practice information was only available to GPs of that practice, but was shared with other GPs if relevant to the discussion or on practitioners' own initiation. The facilitator encouraged participants to ask questions or share remarkable or unexpected aspects of their individual feedback information, facilitated interpretation of the feedback data (eg, by explaining how it is constructed) and probed participants to discuss the consequences of the data for their future daily practice. Each focus group discussion concluded with an exploration of potential issues relevant for follow-up sessions (not reported on in this paper).

The four focus group interactions were transcribed verbatim and anonymised by deleting geographical and personal names. Fragments unintelligible due to simultaneous speech or laughing were transcribed as (unintelligible) and included a timestamp to facilitate fragment location at a later moment of analysis if needed.

## Analysis

Transcripts were analysed using theoretical thematic analysis[31]; audiotapes were consulted to improve interpretation where necessary. Key concepts derived from attribution theory, for example, *external attribution*, formed the initial framework for data coding. MVB first coded one transcript, supplementing the initial codes with codes derived inductively from the data (eg, *gaining information* or *evaluation*). MVB and ND discussed the coding of this first transcript to ensure coding reliability. Applying constant comparison, several codes were modified or merged for code reduction. Next, MVB and ND independently coded a second transcript, adding additional open codes where needed. They discussed their codings in detail until agreement was reached. MVB modified the coding of the first two transcripts accordingly and used the final coding scheme to code the two remaining transcripts. Again, MVB and ND discussed fragments that could not unambiguously be coded until agreement was reached. Although the content of the last focus groups only partly resembled the content of the first two focus groups, the existing codes sufficed to cover the content. This provided evidence for data saturation. As a final step, MVB organised the code themes into a coherent and internally consistent account of what motivates GPs to change.

## Reflexivity

Four members of the research team were medical doctors (MH, ISM, JB, ND), three of whom were practising GPs (MH, ISM, JB). The practising GPs approached ND to initiate the study, induced by their practice experience. They did not participate in data collection or analysis to prevent interaction between their individual experience and the data collection process. One of the focus group facilitators was a medical doctor (not practising). Analysis was primarily done by MVB, who has no medical training and therefore was most distant to the content discussed. This benefited a broad outlook on the data.

van Braak M, *et al. BMJ Open* 2019;**9**:e025286. doi:10.1136/bmjopen-2018-025286

| Box 1 | Focus group O |
| --- | --- |

| 1 | GP A | Well, what I do want to do is check all our cyriax cases [ie, orthopaedic corticosteroid |
| 2 | | injections] to see what the indications were. That is quite a job and it would be very nice if we |
| 3 | | could receive that audit and feedback. But this finding has already provoked me to look that up |
| 4 | | in my own electronic health record to see how many have been done. |
| 5 | CHA | Yes. |
| 6 | GP A | Then we would really have something to talk about, I think. |

## RESULTS

Focus groups were attended by 39 GPs from four regional groups (7 to 10 GPs per focus group). Participants of three focus groups all had practices in an urban area; practices of the fourth focus group's participants were situated in a rural area. GPs' age ranged from 30 to 65 years (the majority being over 50 years of age). Approximately 75 per cent of participating GPs work in a practice together with a partner, with the remainder working alone or in a group practice.

Generally, discussions of audit and feedback items commenced with resolving potential interpretation difficulties. Subsequently, GPs construed an understanding of the information, focusing on probable explanations for deviations from average or between practices. These tended to be followed by GPs expressing their motivations to change.

In the following, we first present aspects of the audit and feedback that contributed to GPs' motivation to change behaviour. We specifically focus on the attributions that GPs used to explain their hesitation to change. Next, we present aspects of the group discussion that contributed to GPs' motivation to change.

### Motivation to change: contribution of audit and feedback information

Two important contributions of audit and feedback information to GPs' motivation to change emerged from our analysis of the focus group discussions. First, the audit and feedback motivated GPs to change *by raising awareness about aspects of their current care practice*. A heightened awareness of one's current practices as reported in the audit and feedback could lead to the realisation that actual care practices differed extensively from perceived care practices or from the norm. This insight frequently induced GPs to express intentions to further reflect on or take additional steps towards practice change, an example of which is presented in box 1.

As this excerpt reveals, being confronted with specific audit and feedback about one's own practice (eg, the finding that the number of cyriax cases, for example a corticosteroid knee injection, deviates considerably from

the norm, line 3) can lead to undertaking specific steps to understand and potentially adapt one's care practices.

Second, the audit and feedback further contributes to GPs' motivation to change *by providing insight into the degree of deviation from norms*. If deviations from 'average' practice are large, changing practice would have a large impact. Minimal deviations from the norm, on the contrary, are judged irrelevant to future practice change. Similarly, the number of patients that are included in a figure signal the impact of potential change. Deviations – either negative or positive – in practice behaviour were less likely a driver for change if only a few patients were involved (box 2).

For these two contributions of audit and feedback to play out, however, GPs pointed out that several conditions have to be met. First, the audit and feedback information *should be reliable and valid*. Suspicion of unreliability of the audit and feedback induces insecurity about possible future actions; one GP said: "it doesn't match with how I feel about it (…), so I don't really know what I should or could do with that". According to another GP, only reliable figures that resemble the GP's own behaviour can rightfully trigger change. Besides being reliable, figures should also be valid. Examples of information that GPs considered to be invalid are: prescriptions that were recorded as prescribed by the GP, but were in fact specialist prescriptions, figures that simply could not be true (eg, only three recorded prescriptions of a medicine that is very commonly prescribed) or drastic changes in particular prescription behaviour from one year to another, while prescription policies were unchanged. If the reader cannot tell what comprises the figures, it remains unclear why increases and decreases in prescriptions, referrals and treatments occur and to whom (or what) these changes can be attributed. As GP A points out in box 3, the data's construction is key to its interpretation.

Perceived limited reliability and clarity frequently induced external attributions, that is, explanations of feedback information by causes external to the GPs influence sphere. If attributed externally, no change talk would follow. As such, unreliability and invalidity of the data compromise the potential contribution of the audit

| Box 2 | Focus group P |
| --- | --- |

| 1 | GP A | There is one other thing that I appreciate about these figures, sometimes deviations are |
| 2 | | enormous but then it's only about ten patients, one isn't going to change policy on that. |

| Box 3 | Focus group V |
|---|---|

| | |
|---|---|
| 1 GP A | But then one soon asks oneself: How did they get to these figures? How is all of this calculated, |
| 2 | if the difference is so large, we haven't started working completely differently a year later. So |
| 3 | there is something there. That does give a lot of interesting information (…). It always comes |
| 4 | down to: What are your norms, why - how are things actually counted? Yes, that's when things |
| 5 | get terribly difficult. That says a lot about the reliability. |
| 6 GP B | I would say they have started to count in a different way. |

and feedback information to GPs motivation to change care practices.

Second, the audit and feedback information *should be specific*. GPs' motivation to change would benefit from broad themes being split up into smaller subthemes (eg, ECGs for specific problems instead of one figure for all ECGs made). The contribution of audit and feedback to GPs' motivation to change would be increased if specific prescriptions or patient information would be available on request. This would help understand, for example, extreme prescription rates or costs (as one GP with high costs wondered: 'Is that because of that Augmentin for that cat bite?'). Having the option to link prescriptions, referrals and costs to specific patients would point GPs at potential behaviour for change.

Third, audit and feedback information *should be recent and recurrent*. Short-term feedback is agreed on as being critical to the effectiveness of audit and feedback as facilitator of change. GPs do not feel the urge to 'learn' from figures that represent their behaviour registered three or more years ago. The feedback should not only be recent, however, but also be recurrent, as box 4 shows.

Fourth, the audit and feedback *should concern GPs' own practices or practices within their own influence sphere*. An example of care practices *outside* the GPs' control are specialist prescriptions. Some GPs suggested to talk to a specialist to discuss deviant figures or refuse particular referrals. More commonly, however, such figures are unlikely subjects for change. GPs would attribute the deviations from 'average' practice represented by these figures to external sources. Examples of such sources are non-GP health professionals (in case of in-hospital treatment) a GP-in-training (whose presence could result in more applications for radiology or lab diagnostics), the practice location (which, for example, might result in fewer home visits if located close to an old-age home or in the care centre), regulations (eg, codes of conduct) and time issues (see box 5).

Despite being potentially problematic, audit and feedback elements that pertain to issues far beyond GPs' control (ie, external attribution) do not induce motivation to change their own practices.

### Motivation to change: contribution of group discussion

In the current study's audit and feedback sessions, the group contributed to GPs' motivation to change in two ways. First, and most importantly, the presence of peers *provided a frame of reference* for interpretation and evaluation of feedback figures. During the focus group discussions, GPs could compare their feedback figures. Comparison can be very informative, as one GP points out in box 6.

Apparently, the need (and motivation) to change practices is more pressing if only one GP's practice deviates from the norm compared with deviation common to all participating GPs. Common deviations are often attributed to demographical or geographical characteristics. Comparing each other's audit and feedback thus functions as a filter, isolating idiosyncratic practice-related variation from region-bound practice variation.

Second, the contribution of group members can *yield important insights that participants would not have been able to achieve on their own*. At times, discussing care practices

| Box 4 | Focus group P |
|---|---|

| | |
|---|---|
| 1 GP A | Those figures are very broad and big, really, so I find it difficult to - if one gets back something |
| 2 | small from your figures, something that you can improve on easily and if one gets back the |
| 3 | figures again after half a year, then I would be more likely to show behaviour change. |

| Box 5 | Focus group OD |
|---|---|

| | |
|---|---|
| 1 GP A | I do tell them quite often to come - make a new appointment, but I do have too few consults |
| 2 | already as well. |
| 3 GP B | Oh, so that doesn't help either. |
| 4 GP A | So that doesn't help either. |
| 5 GP C | We cannot even schedule more consults. |
| 6 GP D | Full is full. |
| 7 GP C | Yes, only if you want to continue working through the evening. |

**Box 6  Focus group OD**

| | | |
|---|---|---|
| 1 | GP A | I think that is interesting, when we see - when I see that I deviate from the national average, and |
| 2 | | we all deviate, then you think: what do we do about that? |
| 3 | GP B | Sure. |
| 4 | GP A | If only mine deviates, then I think: well, I have to do something about that. |

led to an explicitly formulated realisation that particular practices had to be adjusted for reasons proposed during the discussion. In the excerpt displayed in box 7, for instance, one participating GP realised that his hesitation to plan double consults (20 min for patients with multiple complaints) instead of the common 10 min consult (the standard option in Dutch general practice) was unjustified.

Other GPs participating in the interaction in box 7 are used to planning double consults and do not experience the time issues that GP A mentions in lines 1, 2 and 3. The reason behind the differences in the number of double consults planned, GP B poses, is probably understaffing (line 4 and 5). Following this suggestion, GP C and GP D formulate an explicit need for change (lines 8 and 9), and potential objections to the proposed change are warded off in lines 10 and 11. This excerpt shows the benefits of discussing one's audit and feedback with fellow GPs: peers can point out problematical issues or solutions that one has not considered themselves. Also, peers can encourage each other to explore a solution and motivate each other to change – as we see happening in box 7.

Notwithstanding their contributions to GPs' motivation to change, however, group discussions sometimes merely initiated sharing of motivations behind and reflections on practice behaviour without triggering change talk. This type of motivated sharing of best practices occurred quite frequently throughout the group discussions. In general, though, group discussion is perceived to facilitate interpretation of audit and feedback and evaluations of the need for practice change.

## DISCUSSION

The present study qualitatively investigated how audit and feedback group sessions can contribute to GPs' motivation to change practice behaviour to improve care. We framed GPs' responses to the group audit and feedback sessions with attribution theory. This theory contends that an individual's motivation to change behaviour is contingent on their interpretation of the cause behind that behaviour, that is, whether the cause is internal or external, is stable, and is controllable, is central to this approach. Understanding GPs' attributions of behaviour presented during audit and feedback is therefore essential for designing interventions aimed at changing suboptimal care practices.

The presented analysis shows that audit and feedback information can contribute to motivation to change by raising awareness about current practice. At times, that awareness propels GPs to the next step towards change.[20] Audit and feedback can also contribute to motivation to change by providing an indication of the potential impact of change in terms of degree of deviation and number of patients, prescriptions, etc involved. Generally, the lower the impact of change, the lower GPs' motivation to change. A pragmatical consideration seems to be at play here.[23] Even if GPs interpret the behaviour as controllable (something can be done about it), stable (it does not occur randomly) and within their own action range (internal locus), the effort does not outweigh the benefit of change. In these cases, GPs' attributions would not explain the contribution of audit and feedback to GPs' motivation to change. In general, though, the extent to which the audit and feedback pertain to GPs' *individual, controllable and changeable* behaviour is a strong factor in inducing GPs' expressions of change intention – in line with the tenet of attribution theory.[25]

Our findings also point to the key role of *collectively discussing* audit and feedback. As indicated by Trietsch *et al*, social influence and norms affect participants' reflective

**Box 7  Focus group O**

| | | |
|---|---|---|
| 1 | GP A | This is such an eye opener for me, you know? I constantly feel guilty when I plan twenty- |
| 2 | | minute consults twice a day, because then my assistants won't be able to schedule |
| 3 | | enough patients – that's what I think. But I can only do that! |
| 4 | GP B | But darling, we are manning just slightly more patients with twice as many practitioners as you |
| 5 | | have. |
| 6 | GP A | Yes, that's true. |
| 7 | GP B | You know, I mean - that is how it is, really. |
| 8 | GP C | Yes, so you need an extra doctor, and more double consults. |
| 9 | GP A | And a room. |
| 10 | GP C | Yes, and if we charge for the double consults as well, then an extra doctor does not cost a thing. |
| 11 | GP D | Indeed, then you recover the expenses easily. |

behaviour and corresponding intentions to change current practice during peer interaction.[32] Whereas Ivers *et al* conclude that there is very limited evidence for peer-comparison audit and feedback being either more or less effective than individual performance information,[6] our data show that peer comparison in general and group processes in particular stimulate critical appraisal of the audit and feedback and the need for behavioural change.[12 33–35] Peer comparison provides an interpretative framework for individual practice data and peer interactivity provides ample opportunities to explore alternative practices and promising avenues for improvement.[5 10]

Practically, our findings indicate that significant adjustment of contemporary Dutch audit and feedback practices is required to assure prompt and profitable use for improvement of professional practice. To potentially effectuate change, audit and feedback ought to be *reliable, valid, specific, recent* and *recurrent* (cf. recommendations in similar and other research contexts[7–9 11]). We suggest to employ already accessible practice-related information (eg, in the electronic health record) as a starting point for an informative, easily accessible and adaptively employable application serving improvement of GPs' professional practice. Additionally, frequent meetings with GPs practising in the same local or regional area beyond the one currently investigated would be very valuable to facilitate group reflection and discussion across the country. This would promote self-governance of the Dutch GPs, in appreciation of the common needs that led to the Optimal care-Daring doctors movement that started this audit and feedback initiative. Future evaluation research on such interventions would be essential to ensure progressive refinement of the intervention.

By conducting focus group discussions based on personal and comparative audit and feedback, we were able to tap into real-time communicated reflective processes and sketch a comprehensive image of the diversity of attributions and factors impacting participating GPs' motivation to change. Yet, the image could be confounded in three ways. The peer group setting, despite being beneficial to change motivation, might have induced participants to want to look their best. Besides, expressed motivation to change is no guarantee for actual change.[22 36] Therefore, future work exploring the effects of audit and feedback in terms of patient outcomes and compliance with desired practice[6] – both in the short-term and the long-term – is crucial. Finally, the use of a specific type of audit and feedback with GP groups who share an interest in change management processes demands cautious interpretations in terms of transferability to other audit and feedback tools and other GP groups. Yet, the current study's attributional perspective on audit and feedback has certainly enriched our understanding of the complexities of those processes that jointly foster improvement of GP professional practice: individual reflection and critical discussion.

**Correction notice** This article has been corrected since it first published online.

The open access licence type has been amended.

**Contributors** All authors contributed substantially to the conception, design or execution of the reported study. MH and ISM, in collaboration with ND, took initiative for the study. ND, MV, MH and ISM designed the study and collected the data. JB participated in the interpretation of the data. MVB, ND and MV participated in data analysis. MVB was responsible for the write-up of the study. All authors critically revised its content and provided final approval of the version to be published.

**Funding** The authors have not declared a specific grant for this research from any funding agency in the public, commercial or not-for-profit sectors.

**Competing interests** None declared.

**Patient consent for publication** Not required.

**Provenance and peer review** Not commissioned; externally peer reviewed.

**Data sharing statement** The data for this paper may be obtained from the authors upon request.

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
