## [Reviewer comments · BMJ Open]

ARTICLE DETAILS

TITLE (PROVISIONAL)	What motivates General Practitioners to change practice behavior? A qualitative study of audit and feedback group sessions in Dutch General Practice
AUTHORS	van Braak, Marije; Visser, Mechteld; Holtrop, Marije; Stadius Muller, Ilona; Bont, Jettie; van Dijk, Nynke

VERSION 1 - REVIEW

REVIEWER	Terrence McDonald, MD, MSc, CCFP (SEM), Dip. Sport Med Department of Family Medicine University of Calgary Calgary, Alberta Canada
REVIEW RETURNED	19-Sep-2018

GENERAL COMMENTS	Overall, this is a very important topic when considering the medical doctors are part of self-governed profession and peer-review (formal or informal) is considered an important component of this. When reading the abstract and introduction I found it very difficult to discern what question was being addressed and the contextual factors involved. Are Dutch physicians historically less prudent to be 'early adopters' in a change process to improve their pattern of practice? It may assist the readers of this work to be given additional background to the issues faced in the local context and how the data for the mirror process has been historically viewed and interpreted by GPs - which might also address any potential biases affecting the study's outcomes. Throughout the manuscript draft there are numerous grammatical and sentence-structures as well as word choice, that might benefit from further review and revision to either 'soften' the tone or enhance and clarify the argument. For ex: Line 29 Abstract "provoked" Page 7 Line 13 "striking" Page 12 Line 8 "Norm deviations"
--

	Page 15 Line 45 "...GPs' feeling of urgency". Page 17 Lines 15-22 Is 'one' sentence. This needs to be shortened. In the introduction the 'first line' is quite a statement all encompassing and lacks objectivity. The premise of a physicians' - 'personal' decisions needs to be further clarified as it is ambiguous when it might be better to use clinical experience or clinical judgement. The concept of 'motivation' might also be further enhanced if the terms 'intrinsic' vs. 'extrinsic' are used, it is more commonplace for their usage I suspect, citing further from the literature on this might also be helpful. The objectives do not appear to completely match the outcomes, although it is noted in the 'strengths and limitations' sections that the focus was on "intended" not "actual" practice change. More importantly, is it the 'small group' interaction or the content of 'mirror information' itself that is of most interest here? If the latter is the case, then further detail and local context and history need to be included to better establish the case for this particular objective of the study. The Methods section is quite detailed and can be summarized more succinctly in 2 paragraphs, much of this information can be placed into an appendix/additonal documents sections - for inspired readers to review. The Ethics needs its own section or a separate paragraph. Lines 30-41 likely are better placed in the introduction. Line 13-22 also may be better placed in the introduction or under a separate section "Data" -- this is very important information and as reader, I wanted to know this much earlier. The Results section may benefit from a diagram, infographic to truly highly the themes collected/analyzed - shortening this section w/o the dialogue boxes is also a consideration - BUT, highlighting some pertinent findings from the the recordings would be quite helpful for the reader. A reiteration of the 'theoretical framework' particularly the three causal dimensions in the discussion is critical - as it was the basis for the study design - it will strengthen the findings, arguments being made. If demographic data on the participants - is available, it would be very beneficial to consider the differences - particular in regards to age/sex of the provider in how data was viewed and exchanges/feedback from peers was received. Other related research from other countries might be of benefit to highlight how these results might fit, or not -- were they unexpected?
--	---

REVIEWER	Noah Ivers Women's College Hospital - University of Toronto, Canada
REVIEW RETURNED	14-Oct-2018

GENERAL COMMENTS	Thank you for the opportunity to review this interesting paper that explores how Dutch primary care physicians respond to 'mirror information'.
---

The topic is certainly of interest. Initiatives involving the provision of quality indicators to individual clinicians or groups are becoming nearly ubiquitous. This can be associated with small or large intended (and unintended) effects. It is important to understand how and why the recipients of such data respond to be able to refine these interventions and meet shared goals. The findings have the potential to inform practice in the Dutch context in terms of identifying ways to improve the reports, the data they are based upon, and the way they are delivered.

I have some concerns about the applicability of this article and its findings for a broad readership. I hope the following suggestions are helpful in that regard.

- 1) I did not see a relevant reporting checklist (e.g. COREQ). This may have been useful... Reflexivity was not addressed per se. Details of participants belongs usually in results not in methods section
- 2) This topic - the use of data and the discussion of data amongst peers to spur clinical performance improvement actually has quite a large literature that was not considered - it is generally described in the literature as 'audit and feedback' although there are many synonyms. See a recent 2018 article in Implementation Science on this very topic of providing data to groups of clinicians out of Alberta Canada. See also the article in Annals of Internal Medicine in 2016 led by Brehaut on practice feedback and the citations within.
- 3) Relatedly, there is the Feedback Intervention Theory and other relevant theories - many of which are cited in the Cochrane review on the topic of audit and feedback referenced in this paper - and it is not clear if the 'Attribution Theory' was chosen or how it fits with more commonly used theories and models for this field. It is good to take a new perspective, but if one does not try to fit the findings with and/or build upon extant knowledge, we risk confusing the science rather than progressing it.
- 4) Similar to my above point, I'm not certain that grounded theory was the best choice here, given that a) there is a great deal known already on the topic and b) you seem to have also chosen an a priori theory that guided the work. I wonder if maybe you used thematic/content analysis and a constant comparative methodology rather than true grounded theory and if so, that might be more appropriate in my opinion.
- 5) The intervention itself is inadequately described. What were the quality indicators, why were those chosen, etc.
- 6) some of the findings and in particular the points made in the discussion, while seemingly sensible, could possibly be better supported by the data presented
- 7) I have some concern with focusing on motivation/intention without emphasis on the intention-behaviour gap and without discussion of how the self-reported nature of the intentions are at risk of bias, especially in a focus group setting, where there may be a desire to 'put ones best foot forward'. Another limitation that could be further emphasized is the small sample here. I struggle to believe that saturation was achieved with this sample size or that a great enough effort was made to search out dissenting views and incorporate them into the findings. Some member checking or other approach to confirm the findings might have been useful.
- 8) It might be useful to focus the results/discussion a bit, especially given the smaller sample size. I think this might help alleviate concerns about saturation. I wonder if insights on how

	best to organize these group discussions may, like the Alberta Canada paper mentioned above, may be a way that this paper, if a little more focused, could be quite interesting.
--	--

VERSION 1 – AUTHOR RESPONSE

- Dr. McDonald commented that it was difficult to discern the manuscript's main question and contextual factors involved from the abstract and introduction. We think that the formulation of the main and sub research questions might have contributed to the confusion and have now reformulated them to more clearly convey the manuscript's focus, see abstract under 'Objectives' and the last paragraph of the Introduction. The contextual factors now receive additional attention in the paragraph describing the local context (Introduction, see also 'Setting' in the Methods section).
- Dr. McDonald asks whether Dutch physicians are historically less prudent to be 'early adopters' in a change process to improve their pattern of practice. We suspect this question might have been raised by the way the results are framed in the abstract as well as the lack of background information on the issues faced in the local context. Dutch GPs are far from resistant against change, as we now describe in the Introduction (paragraph 5). The additional information on the local context also responds to dr. McDonald's call to establish more firmly the case for the second objective of the study (to determine how group discussion in these audit and feedback sessions contributed to GPs' motivation to change).
- Dr. McDonald commented that the Introduction's first line "is quite a statement all encompassing and lacks objectivity". To deal with this concern, we have now adjusted the line into 'In taking the Affirming the Hippocratic oath, all doctors, including general practitioners – like other doctors – express their intention, to treat patients to the best of their ability.'
- With 'personal' choices, which dr. McDonald observed to need further clarification, we hint at situations where there is more than one solution. In these situations, some GPs might tend to opt for solution A while others prefer solution B. As preference was already included in the list of factors, we left 'personal choices' out. As we think 'clinical experience', which dr. McDonald mentioned in his comment, is an important factor as well, we have added this to the list.
- Similarly, we disambiguated the next sentence by replacing 'personal decisions based on evidence, costs and patient satisfaction' with 'clinical judgment based on considerations of evidence, clinical experience and patient preferences'.
- Referring to the concept of motivation, dr. McDonald suggests that it might be helpful to discuss the terms 'intrinsic' and 'extrinsic' motivation. Although we do agree that this distinction is used a lot in the literature on motivation, our focus is not so much on classifying the type of motivation that GPs might have, but on the attributions that might affect their motivation: what motivates them, and why are they motivated by it? Despite its ubiquitous presence in the motivation literature, therefore, we choose not to refer to the intrinsic-extrinsic distinction.
- We thank dr. Ivers for his suggestion to include literature on audit and feedback in our manuscript, especially the specific articles mentioned. We have now enriched the Introduction section with additional mentions on audit and feedback research, including Cooke et al. (2018) and Brehaut et al. (2016).
- To connect the current research more firmly to the existing audit and feedback literature, we also decided to replace the local term 'mirror information' with the commonly used term 'audit and feedback' throughout the manuscript.

- Dr. Ivers comments on the absence of a clear link between the new perspective we take in this manuscript and other relevant theories on audit and feedback. We understand that acknowledging extant knowledge is important and now refer to these theories in the Introduction, paragraph 3. At the same time, however, we contend that taking an attributional perspective has an important contribution to make to the science of audit and feedback. As the relevance of this perspective might not have been clear from the original manuscript, we now addressed this issue more explicitly in the third and fourth paragraph of the Introduction.

- We revised the method section following both reviewers' helpful suggestions:

- We placed information on ethics in a separate paragraph.

- The background information about the Dutch healthcare system and the specific form of audit and feedback data have been reorganized under the subheader 'Setting' in the methods section.

- We moved participant details from the methods to results section.

- The intervention itself is now described in more detail, including the topics discussed and the reasons for topic selection.

- We added a subsection on reflexivity.

- We attempted to dense the sections on data collection and analysis for readability.

- We thank dr. Ivers for his useful considerations of the suitability of grounded theory as our method of analysis. As we mentioned in our manuscript, we analyzed the transcripts using "principles of Grounded Theory" and used a priori codes derived from Attribution Theory, but we did not aim to develop a theory. Although Grounded Theory has been used as such in previous literature (Watling & Lingard, 2012; Bryant, 2002), we agree with dr. Ivers that our analysis would indeed better be regarded an application of thematic analysis instead of true Grounded Theory. We have adapted the analysis section of the Methods accordingly.

- In response to dr. Ivers' comment on the inadequacy of our intervention description, we have detailed our description of the (choice of) quality indicators.

- Upon the editor's request, we have added a 'Patient and Public Involvement' statement to the Methods section.

- Upon the editor's request, we now cite all boxes in the main text.

- As dr. McDonald noted, the objectives and aim of the study did not completely match the results. The study's dual focus on the content of the mirror information and the small group interaction was not clear from the organization of the results section, which might have contributed to the result section's lack of focus that dr. Ivers commented on. We have now adapted this section such that the results are structured according to the sub questions guiding our research. Some factors that were originally presented as motivations to (not) change practice or as factors that would increase motivation to change should probably be better understood as conditions under which the audit and feedback can contribute to GPs motivation to change. We have now denoted them as such (e.g. under the section 'Motivation to change: contribution of audit and feedback information' from paragraph 3 onwards).

- (For a reply to dr. McDonald's comment that further detail and local context and history need to be included to better establish the case for the study's second objective: see the Introduction replies.)

- Dr. McDonalds suggested that the Results section may benefit from an infographic to highlight the main themes. Since we have now arranged this section more conveniently, we think that this

suggestion has now become superfluous. Instead, we densified the results section. We left out the box that was originally numbered Box 5; this box served to explain a point that was also illustrated with the excerpt presented in the box originally numbered Box 4. Furthermore, we removed the box originally numbered Box 6, as its relation to the contribution of the audit and feedback information to motivation to change was less clear. In this way, we have brought more focus to our results section, as suggested by Dr. Ivers.

- Dr. McDonalds suggested to reiterate part of the theoretical framework in the discussion to strengthen the findings and argumentation. We now start our discussion with a short recapitalization of Attribution Theory to facilitate our discussion of the findings in light of this theory.

- Dr. McDonalds proposes to consider how participants of various age/sex respond to the data and the exchange with peers. Although we agree that exploring participants' responses in relation to demographic variables might be of interest, our sample size is limited and would therefore not allow for systematic analysis. For that reason, we chose not to conduct such analysis.

- Dr. Ivers commented that some findings and points made in the discussion could possibly be better supported by the data presented. We think this point also relates to the correspondence between the questions asked and data presented, which was sometimes unclear in the original manuscript. To alleviate this concern, we reformulated parts of the results to stay closer to the tenor of the data and adapted the relevant sections of the discussion (current second, third, and fourth paragraph).

- Dr. Ivers expressed some concerns about our discussion of motivation without emphasis on the intention-behaviour gap and the self-reported nature of the motivations. We understand his concerns and agree that the original manuscript addressed these issues only to a limited extent. We now take up this issue more explicitly in paragraph six of the Discussion section. In this paragraph, we also mention the member checking we conducted to ensure the recognizability and usefulness of the study's findings.

- As suggested by the editor, we revised the manuscript title to include the research question, study design, and setting. It now reads: 'What motivates General Practitioners to change practice behavior? A qualitative study of audit and feedback group sessions in Dutch General Practice'.

- As suggested by the Editor, we have included in our submission a completed copy of the SRQR checklist indicating the manuscript's line/page number where the relevant information can be found.

- Dr. McDonalds mentioned several phrases that might need revision to soften the tone or clarify the argument. We reformulated those phrases still included in the manuscript (striking, norm deviations, last paragraph of Discussion section) to that end.

- Dr. Ivers expressed some concerns about the applicability of the article and its findings for a broad readership. We now explicitly describe our results in terms of features of the audit and feedback intervention, which makes the study's findings applicable to other interventions with similar features.

On a general note, we would like to mention that following the suggestions of the reviewers resulted in extra text in some places and less text in others. We have tried to limit our word count as much as possible (but due to the qualitative nature of the results, we could not bring it back to less than 4126 - similar to our first submission).

VERSION 2 – REVIEW

REVIEWER	Terrence McDonald, MD, MSc, CCFP (SEM), Dip. Sport Med Department of Family Medicine University of Calgary
REVIEW RETURNED	31-Dec-2018

GENERAL COMMENTS	p. 9 Lines 56-58 require more clarity on the 'two' contributions, the two paragraphs that follow (on p. 10) need to be tied in further to these contributions. Suggest clarification of 'cyriax cases' in Box 1 and 'cyriax operations' in the lines that follow Box 1; although perhaps conventional in the Dutch medical context it is not a common term in medical/surgical reports or common medical dialogue in the Canadian context. Suggest a brief clarification to include 'orthopaedic surgeries', or 'orthopaedic diagnosis' or 'musculoskeletal assessments'. p. 11 Line 37-43 is a concise clear example about: treatment of a 'cat bite', it much more clear and the point is more discernible from a medical terminology perspective. Box 2 - Perhaps consider clarifying 'what type' of information presented - was it 'positive' or 'negative' in nature for the GP(s) to lead to the dismissal of a small variance. The reader might be also be left wondering what was 'valid' about the information presented, consider a link to a 'summary' of the data presented if my is believed to add to the point being made. p. 12 Line 9-26. Very lengthy paragraph, consider omitted 1-2 points, or create 2 shorter sentence, specificity to clarify lines on "...other GP's", consider a again shortening a make the point more concisely. p. 12 line 51, Should read..."First, and most importantly..." 'Discussion' - line 24-28 need to be re-worded following the sentence ending with Attribution Theory. Suggest..."This theory contends that a participants motivation to change etc.... Line 33 non-optimal vs. ? suboptimal. Line 47"at play here". Consider a full-stop here to end sentence them continue with the points that follow as a 2nd sentence. P. 15 line 7-8 suggest alternate word choice 'confounded' vs. 'blurred'. p. 15 line 58 'detailed understanding' is quite a bold statement, given the small sample size of like-minded GPs (likely early adopters of new and innovations for change management processes, this needs to be highlighted as a possible weakness of the work, it may be skewed and perhaps not generalizable. under term: "professionalization" - needs clarification and tie in with standards that are upheld by the medical profession and/or medical authority or association. Overall the discussion will benefit from several items and will strengthen the paper's main findings that are otherwise very valuable for this form of feedback and audit. Consider fort the 'Discussion':
---

	1) To be shorter (3 concise paragraphs, 4 max.), it appears to start off relatively well with restating the theory (s) employed, and states some of the results, but then seems to become unfocused with some elaborate language that may not apply (line 58 p. 15). 2) Indicate how this adds to literature - which is mostly accomplished, but could be shortened, theoretical literature vs. similar research work completed in this field. 3) Indicate how it can be applied and re-worked with future research with specific examples in the Dutch context -- where several sources of clinical/practice/insurance feedback already seem to be offered to GPs. Consider: What is the overall 'direction' that the governance of Dutch GPs is interested in given that this research was borne out of a desire from a group of like-minded GPs who wanted this type of study (and mirror interview feedback)? This was stated early in the paper, but could be reiterated as it is key. Consider the concept of 'self-governance' vs. 'external governance' (government, public agencies, insurance agencies) --- to strengthen the argument being made in lieu of the findings this important work.
--	---

VERSION 2 – AUTHOR RESPONSE

- Dr. McDonald noted that 'two contributions' discussed under 'Motivation to change: contribution of audit and feedback information' required more clarity. Although it might not have been clear that 'raising awareness' is discussed as the first contribution and 'providing indications of possible impact of change' is discussed as the second, we contend that our discussion of both points both explains and illustrates these respective contributions of the actual mirror and feedback information to GPs' motivation to change. We have slightly altered the text to clearly signal each contribution.

- We thank Dr. McDonald for his suggestion to clarify 'cyriax cases' in Box 1 and the lines that describe Box 1. We have added "[i.e., orthopaedic corticosteroid injections]" in line 1 in Box 1 and "for example a corticosteroid knee injection" in the second line following Box 1.

- As suggested, we have now clarified the type of information that GPs dismiss as trivial practice behavior variation (see lines directly above Box 2).

- As the type of information presented in the audit and feedback is very diverse, we think that a summary of the data beyond the description provided under Methods (Setting > paragraph 2) would not be very helpful in solving the readers' questions about the validity of the data. Yet, we agree with Dr. McDonald that what was 'valid' about the information presented needs some further elaboration. Therefore, we have now added some examples of situations/figures which the GPs in our focus groups considered invalid.

Following Dr. McDonald's suggestion, we have shortened the paragraph following Box 4 by omitting two examples of external sources. We have also created two separate sentences from the last sentence, which originally started with "GPs would attribute...". Both changes contribute to the conciseness of the paragraph.

- As recommended, we have changed "Most importantly," in the second line under 'Motivation to change: contribution of group discussion' into "First, and most importantly,".

- As suggested, we have reworded the third and fourth sentence under Discussion into "This theory contends that an individual's motivation to change behavior is contingent on their interpretation of the cause behind that behavior, i.e. whether the cause is internal or external, is stable, and is controllable."

- Similarly, we changed non-optimal (in the last sentence of the first Discussion paragraph) into “suboptimal” and split one long sentence in two shorter ones in the second Discussion paragraph.

- As suggested, we have changed ‘blurred’ in the second paragraph on pg. 16 into ‘confounded’.

- We took into consideration Dr. McDonalds comment that ‘detailed understanding’ (same paragraph) is quite a bold statement given the possible limits to the generalizability of the findings to other GP groups. We still think that the current analysis indeed gives a detailed understanding of these particular GP groups’ change considerations, although findings in other GP groups might differ due to various reasons. We have now included these considerations in the text: “Finally, the use of a specific type of audit and feedback with GP groups who share an interest in change management processes demands cautious interpretations in terms of generalizability to other audit and feedback tools and other GP groups.” (last paragraph Discussion)

- Dr. McDonald recommended to clarify the term ‘professionalization’ (last paragraph Discussion) to make it tie in with profession standards. Following his suggestion, we have adopted Ivers et al. (2012)’s terminology of describing effectiveness measures. Using the terms employed in this review, we now talk about “effectiveness of this audit and feedback tool in terms of patient outcomes and compliance with desired practice” and “improvement of GP professional practice”.

- We have made several alterations to the Discussion section in response to Dr. McDonald’s recommendations for making this section more concise: 1. We have deleted numerous bits of text that we thought might have contributed to the text becoming “unfocused with some elaborate language that may not apply”. 2. We have retained our current references to theoretical literature and similar research work, but at the same time tried to integrate them more in the text. 3. Although we managed to shorten the text into fewer paragraphs, we believe that the currently provided information is too dense and varied to be comprised into three or four paragraphs.

- Dr. McDonald further suggested to indicate how the intervention can be applied and re-worked with future research. We have now slightly elaborated on the example about the Electronic Health Record, which we think is a clear example of a way to adapt current resources to fit the specific needs and possibilities of GPs in the context of professional practice improvement.

- We have now also included Dr. McDonalds recommendation to revisit in the Discussion the movement that initiated this audit and feedback intervention (second-last paragraph).

- Finally, a note on the reporting guidelines. Since we uploaded the reporting guidelines for the last revision and only slight changes were made, we refer to that revision’s reporting guidelines document.

VERSION 3 – REVIEW

REVIEWER	Terrence McDonald, MD, MSc, CCFP (SEM), Dip. Sport Med Department of Family Medicine University of Calgary Calgary, Alberta Canada
REVIEW RETURNED	18-Mar-2019

GENERAL COMMENTS	First, congratulations to the authorship team on refining this work - it is excellent and will likely have an impact both in this area of work and within practice application(s). A few minor edits for your consideration: Pg. 5 2nd Paragraph: lower case 'a' for: ...an attributional.
--

3rd Paragraph: 4th Line: Consider: "In the last decade" ('since' invokes some 'vagueness' which may not fit for your point, might be strengthened with the above)..., then state GPs have started to use..."

Pg. 6

2nd Paragraph: Considering making 2 sentences - strengthen your point and keep your reader 'focused'. It otherwise reads as a 'run-on' sentence.

Similarly:

Pg. 8

Sentence starting with "Focus groups"...consider 2 sentences, for the same reason as noted above.

Good luck with your submission and ongoing work.
TM